# Tuberculosis preventive therapy for people living with HIV: A systematic review and network meta-analysis

Mercedes Yanes-Lane[1], Edgar Ortiz-Brizuela[1,2], Jonathon R. Campbell[1,3], Andrea Benedetti[1,3,4], Gavin Churchyard[5,6], Olivia Oxlade[1], Dick Menzies[1,3]*

1 Respiratory Epidemiology and Clinical Research Unit, McGill International TB Centre, McGill University, Montréal, Québec, Canada, 2 Department of Medicine, Instituto Nacional de Ciencias Médicas y Nutrición Salvador Zubirán, Mexico City, Mexico, 3 Department of Epidemiology, Biostatistics and Occupational Health, McGill University, Montreal, Canada, 4 Department of Medicine, McGill University, Montreal, Canada, 5 The Aurum Institute, Parktown, South Africa, 6 School of Public Health, University of Witwatersrand, Johannesburg, South Africa

* dick.menzies@mcgill.ca

**Data Availability Statement:** All relevant data are within the manuscript and its Supporting Information files.

## Abstract

### Background

Tuberculosis (TB) preventive therapy (TPT) is an essential component of care for people living with HIV (PLHIV). We compared efficacy, safety, completion, and drug-resistant TB risk for currently recommended TPT regimens through a systematic review and network meta-analysis (NMA) of randomized trials.

### Methods and findings

We searched MEDLINE, Embase, and the Cochrane Library from inception through June 9, 2020 for randomized controlled trials (RCTs) comparing 2 or more TPT regimens (or placebo/no treatment) in PLHIV. Two independent reviewers evaluated eligibility, extracted data, and assessed the risk of bias. We grouped TPT strategies as follows: placebo/no treatment, 6 to 12 months of isoniazid, 24 to 72 months of isoniazid, and rifamycin-containing regimens. A frequentist NMA (using graph theory) was carried out for the outcomes of development of TB disease, all-cause mortality, and grade 3 or worse hepatotoxicity. For other outcomes, graphical descriptions or traditional pairwise meta-analyses were carried out as appropriate. The potential role of confounding variables for TB disease and all-cause mortality was assessed through stratified analyses.

A total of 6,466 unique studies were screened, and 157 full texts were assessed for eligibility. Of these, 20 studies (reporting 16 randomized trials) were included. The median sample size was 616 (interquartile range [IQR], 317 to 1,892). Eight were conducted in Africa, 3 in Europe, 3 in the Americas, and 2 included sites in multiple continents. According to the NMA, 6 to 12 months of isoniazid were no more efficacious in preventing microbiologically confirmed TB than rifamycin-containing regimens (incidence rate ratio [IRR] 1.0, 95% CI 0.8 to 1.4, $p = 0.8$); however, 6 to 12 months of isoniazid were associated with a higher incidence of all-cause mortality (IRR 1.6, 95% CI 1.2 to 2.0, $p = 0.02$) and a higher risk of grade

**Funding:** This work was funded by the Bill & Melinda Gates Foundation (Grant Number INV-003634, received by DM). The initial study questions for the papers included in the PLOS Collection were drafted together with input from staff of the Bill & Melinda Gates Foundation, but they had no further role in study design, data collection and analysis, decision to publish, or preparation of the manuscript.

**Competing interests:** The authors have declared that no competing interests exist.

**Abbreviations:** ALT, alanine transaminase; ART, antiretroviral therapy; AST, aspartate transaminase; CDC, Centers for Disease Control and Prevention; IGRA, interferon gamma release assay; IQR, interquartile range; IRR, incidence rate ratio; LTBI, latent tuberculosis infection; NMA, network meta-analysis; PLHIV, people living with HIV; PRISMA, Preferred Reporting Items for Systematic Reviews and Meta-Analyses; RCT, randomized controlled trial; RD, risk difference; TB, tuberculosis; TPT, tuberculosis preventive therapy; TST, tuberculin skin test; WHO, World Health Organization.

3 or higher hepatotoxicity (risk difference [RD] 8.9, 95% CI 2.8 to 14.9, $p = 0.004$). Finally, shorter regimens were associated with higher completion rates relative to longer regimens, and we did not find statistically significant differences in the risk of drug-resistant TB between regimens. Study limitations include potential confounding due to differences in posttreatment follow-up time and TB incidence in the study setting on the estimates of incidence of TB or all-cause mortality, as well as an underrepresentation of pregnant women and children.

## Conclusions

Rifamycin-containing regimens appear safer and at least as effective as isoniazid regimens in preventing TB and death and should be considered part of routine care in PLHIV. Knowledge gaps remain as to which specific rifamycin-containing regimen provides the optimal balance of efficacy, completion, and safety.

## Author summary

### Why was this study done?

- People living with HIV (PLHIV) have a high risk of developing and dying of tuberculosis (TB) disease.

- There are several tuberculosis preventive therapy (TPT) regimens available for PLHIV (i.e., 6, 12, and 36 months of isoniazid (6H, 12H, and 36H), 3 months of isoniazid plus rifampin (3HR), and 1 or 3 months of isoniazid plus rifapentine (1HP and 3HP). To date, there are no direct comparisons between these preventive treatments.

### What did the researchers do and find?

- We performed a systematic review and network meta-analysis (NMA) of studies comparing 2 or more of the currently recommended TPT regimens among PLHIV.

- The NMA approach allowed us to use data from published randomized controlled trials (RCTs) to indirectly compare different TPT strategies.

- Shorter, rifamycin-containing regimens (3HR, 3HP, and 1HP) had better completion rates than isoniazid-based regimens (6H, 12H, and 36H).

- When compared to rifamycin-containing regimens, isoniazid-based regimens were associated with a higher mortality risk and a higher frequency of severe adverse events.

- Rifamycin-containing regimens were as good as isoniazid-based regimens at preventing TB disease, and there was no evidence that they contributed to development of drug resistance.

**What do these findings mean?**

- Rifamycin-containing regimens are as safe and effective as isoniazid-based regimens in PLHIV and should be considered part of routine care.

- Further research is needed to define which rifamycin-containing regimen is most beneficial in PLHIV.

## Introduction

Tuberculosis (TB) is the leading cause of death among people living with HIV (PLHIV) [1]. In 2019, approximately 208,000 people died of HIV-associated TB [2]. Tuberculosis preventive therapy (TPT), with or without antiretroviral therapy (ART), has proven to reduce progression to active TB and mortality among PLHIV [3]. Previously, the World Health Organization (WHO) considered using either 6 or 36 months of isoniazid (hereafter, 6H and 36H, respectively) as the preferred TPT regimens, based on the setting's TB incidence [4]. However, significant disadvantages of isoniazid-based regimens, including their length, low completion rates, and potential for hepatotoxicity [5], led to the search of better completed and more tolerable rifamycin-containing regimens [6–11].

In light of this new evidence, the WHO updated their guidelines in 2020 to include the use of 3 months of daily isoniazid plus rifampicin (3HR) and 3 months of weekly isoniazid plus rifapentine (3HP) as the preferred TPT regimens along with 6-9H, regardless of HIV status [12]. One month of daily isoniazid plus rifapentine (1HP) and 4 months of daily rifampin (4R) were considered alternative regimens [12]. Nevertheless, caution is advised when prescribing rifamycin-containing regimens to PLHIV due to the risk of drug–drug interactions, for example, between rifamycins and protease inhibitors (e.g., darunavir) [12,13]. Finally, 36H is still recommended for PLHIV in settings with a high risk of TB transmission, in view of the considerable risk of reinfection [12].

According to previous network meta-analyses (NMAs) of latent tuberculosis infection (LTBI) treatment regimens, both isoniazid-based and rifamycin-containing regimens are effective in preventing TB disease among PLHIV and other risk groups [14]. However, there are few direct head-to-head comparisons between isoniazid-based and rifamycin-containing regimens among PLHIV. Hence, we aimed to assess the effectiveness and safety of different TPT regimens among PLHIV through a systematic review and NMA of clinical trials comparing a TPT regimen with either placebo, no treatment, or other treatment regimens.

## Methods

### Data sources and searches

This systematic review adheres to the Preferred Reporting Items for Systematic Reviews and Meta-Analyses (PRISMA) for NMA guidelines (PRISMA in S1 File), and our protocol was prospectively registered with PROSPERO (CRD42020177338). Given the dearth of methods available to control for heterogeneity introduced by observational studies in NMAs, a decision was made to only include randomized trials in our study. Protocol amendments to our initial submission to PROSPERO were carried out accordingly.

We designed a search strategy to identify all randomized controlled trials (RCTs) comparing at least 1 TPT regimen to placebo, no treatment, or other TPT regimens among PLHIV, regardless of age, setting, and baseline tuberculin skin test (TST) or interferon gamma release assay (IGRA) status. At least 1 of the following outcomes must have been reported: efficacy (in terms of incidence of microbiologically confirmed active TB or clinically diagnosed TB and all-cause mortality), grade 3 or worse hepatotoxicity [15], treatment completion, or occurrence of drug-resistant TB after treatment. We searched MEDLINE, Embase, and the Cochrane Library to identify all clinical trials published from database inception until June 9, 2020. The complete search strategy is available in Material A in S1 File. No additional articles were identified from the retrieved articles' references or relevant systematic reviews identified during the search.

## Study selection

All titles, abstracts, and full texts were independently assessed by 2 reviewers (EOB and MYL). Studies were screened without language restriction. Included studies compared at least 1 regimen of interest to placebo, no treatment, or to another TPT regimen. Regimens of interest were those recommended in guidelines published in 2020 by the Centers for Disease Control and Prevention (CDC) and WHO guidelines (i.e., 3 to 4R, 3 to 4 HR, 3HP or 1HP) as well as different durations of daily isoniazid [12,13]. With the exception of 3HP, intermittent TPT regimens were excluded, as well as regimens containing ethambutol or pyrazinamide, since they are no longer recommended [4]. Moreover, included studies must have presented results stratified by HIV status or exclusively performed among PLHIV. Finally, only the following study designs were considered: RCTs, including factorial or parallel RCTs.

Studies were excluded if they did not compare 2 or more TPT regimens of interest or at least 1 TPT regimen of interest and either placebo, no treatment, or another regimen not of interest that would permit indirect comparisons of any of the currently recommended regimens. For studies in languages other than English, members of the team fluent in the language were asked to asses the eligibility of the study.

## Data extraction and quality assessment

Two independent reviewers (EOB and MYL) extracted data from all included studies using a predefined extraction form (Table A in S1 File). Disagreements were resolved by consensus with a third reviewer (DM). If studies reported additional details, such as long-term follow-up in another manuscript, information was extracted from both.

The quality of the included studies was evaluated using criteria from the revised Cochrane risk-of-bias tool for randomized trials (RoB 2) that was adapted for simplicity (Table B in S1 File) [16]. One or more items from each RoB 2 domain were included based on the intention-to-treat effect. The following 5 areas were assessed: bias arising from the randomization process, bias due to deviations from intended interventions, bias in measurement of the outcome, bias in missing outcome data, and bias in the selection of the reported results. Studies were categorized as with a low risk of bias when there was a low risk of bias in the randomization process and in at least 2 of the 4 remaining domains. If these conditions were not met, the study was considered at high risk of bias. Two independent reviewers (EOB and MYL) conducted the quality assessment, and any disagreements were resolved by consensus.

## Outcomes

For information on outcomes, only intention-to-treat data were included. For studies where long-term follow-up was available (and met our inclusion criteria), information from these

was used in the analysis. As explained above, 5 main outcomes were assessed: incidence of microbiologically confirmed TB, microbiologically confirmed and clinically diagnosed TB, all-cause mortality, grade 3 or worse hepatotoxicity, treatment completion, and drug-resistant TB occurrence after therapy.

## Definitions

We defined microbiologically confirmed TB as a diagnosis made during treatment or post-treatment follow-up using acid-fast bacilli smear, culture, and/or molecular tests (e.g., GeneXpert). Clinically diagnosed TB was defined as a diagnosis made using chest X-ray abnormalities or suggestive histology (positive for necrotizing granulomas). All-cause mortality included outcomes during treatment or posttreatment follow-up. Grade 3 or worse hepatotoxicity was defined as an increase of alanine transaminase (ALT) or aspartate transaminase (AST) of at least 5 times the upper limit of normal according to criteria described elsewhere [17]. Only adverse events detected during the treatment phase were included in this analysis. If the study reported adverse events after treatment completion combined with adverse events during treatment, these results were excluded from the analysis. We used individual study definitions for TPT completion. Drug resistance was classified as isoniazid resistance (resistance to isoniazid with or without resistance to other first-line TB drugs) and rifampin resistance (resistance to rifampin with or without resistance to other first-line TB drugs).

## Data synthesis and statistical analysis

Differences in treatment effects were summarized using incidence rate ratios (IRRs) or risk differences (RDs) per 100 persons randomized, as appropriate. Since previous systematic reviews assessing TPT effectiveness among PLHIV have found a major degree of clinical and methodological heterogeneity, we decided a priori to perform a random-effects NMA using a graph theoretical approach to estimate the pooled IRR or pooled RD for the outcomes of microbiologically confirmed active TB, all-cause mortality, and grade 3 or worse hepatotoxicity [18].

Briefly, the graph theoretical approach uses a frequentist method to calculate NMA effect estimators. It is based on electrical theory in which variance corresponds to resistance, treatment effects to voltage, and weighted treatment effects to current flows [18]. When applied to a pairwise meta-analysis, it can produce both random-effects and fixed-effects models. This method has been found to be equivalent to Bayesian mixed treatment comparisons [19].

Multi-armed studies were included as 2 armed comparisons for all possible treatment combinations, accounting for comparisons belonging to the same study by using identical study labels and adjusting the standard errors accordingly [18]. To test for inconsistency in the NMA, the net heat plot method was used [20]. This method assesses the contribution of each study to inconsistency in the network by temporarily removing each and calculating the differences in inconsistency. It uses colors within a matrix to graphically display the level of inconsistency from each design [20]. Potential indicators of inconsistency were further analyzed by looking at the distribution of variables across studies and identifying potential outliers.

We used a frequentist approach to calculate the average treatment ranking (P-score) [21]. This method uses the point estimates and standard errors of the frequentist NMA to rank treatment estimates. The P-scores should be interpreted as the extent of certainty that one treatment is better than another, averaged over all competing treatments. In our analysis, P-scores closer to 1 indicate a higher certainty.

Given the heterogeneity in definitions of treatment completion between studies, a meta-analysis was not performed for this outcome, and only a descriptive analysis was reported.

Likewise, the low number of studies reporting drug-resistant TB precluded our ability to perform an NMA. Instead, the pooled cumulative incidence of drug-resistant TB was estimated via direct pairwise meta-analysis using a random-effects model. The latter was performed by estimating a generalized linear mixed model with logit transformation, using the total number of patients randomized to the intervention arm as the denominator.

As suggested by the *Cochrane Handbook for Systematic Reviews of Interventions*, treatments were grouped into nodes to maximize similarity of the interventions within each node while minimizing similarity across them [22]. In order to compare all mono-isoniazid regimens against rifamycin-containing regimens, we created a node with all mono-isoniazid regimens (i.e., from 6 to 72 months of isoniazid) and another with all rifamycin-containing regimens (i.e., 3HR, 3HP, and 1HP). After that, we divided mono-isoniazid regimens into 2 different nodes based on treatment duration (i.e., 6 to 12 months of isoniazid and 24 to 72 months of isoniazid). Finally, since pyrazinamide-containing regimes are no longer recommended for TB prevention, no comparisons were presented using this regimen. However, study arms containing pyrazinamide were used for indirect comparisons of the regimens of interest.

### Secondary analysis

In planned secondary analysis, all outcomes were assessed for each individual (i.e., disaggregated) TPT regimens. Moreover, post hoc analyses were carried out to explore the potential effect of TB reinfection on the outcomes of microbiologically confirmed active TB and all-cause mortality. The NMA was stratified by TB incidence rates in the study setting (using a cutoff of 300 cases per 100,000 population) as well as posttreatment follow-up duration (using a cutoff of 1 year of posttreatment follow-up). Additionally, after obtaining our results, we decided to investigate the association of rifamycin-containing regimens with a lower all-cause mortality through a sensitivity analysis excluding studies of rifamycin-containing regimens with >90% use of ART, a stratified analysis of studies in which more, or less than 50% of participants had received ART, and meta-regression, adjusting for proportion of study participants who had ever received ART using the method described by Lumley [23].

Lastly, we conducted a stratified analysis by TST/IGRA status. However, since the number of studies reporting microbiologically confirmed TB stratified by TST or IGRA status was limited, we reported the results of this analysis using the outcome of microbiologically confirmed combined with clinically diagnosed TB. Data on TB incidence in person years (microbiologically confirmed or clinically diagnosed) were abstracted if reported or estimated based on numbers of reported cases and person-years of follow-up calculated from median time of follow-up.

All statistical analyses were carried out using the meta (version 4.15–1) [24] and netmeta (version 1.2–1) [25] packages in R software version 3.6.3 [26].

### Results

We identified a total of 6,466 unique studies, of which 157 were selected for full-text assessment. Of these, 20 studies (reporting 16 different clinical trials) were included [6–11,27–40] (Fig 1). A complete list of the 137 excluded reports after full-text review with reasons for their exclusion is available in Table C in S1 File.

Sixteen reports from 12 clinical trials of TPT regimens among PLHIV were excluded for the following reasons. Four studies from 3 clinical trials were excluded because no regimen of interest was assessed [41–44]. Three clinical trials included only 1 TPT regimen of interest but were excluded because no indirect comparisons were possible [45–47]. Nine studies from 6 clinical trials were excluded for other reasons: 1 trial compared empiric active TB therapy against 6H [48–50]; a second evaluated the safety of isoniazid preventive therapy at 2 timings during pregnancy

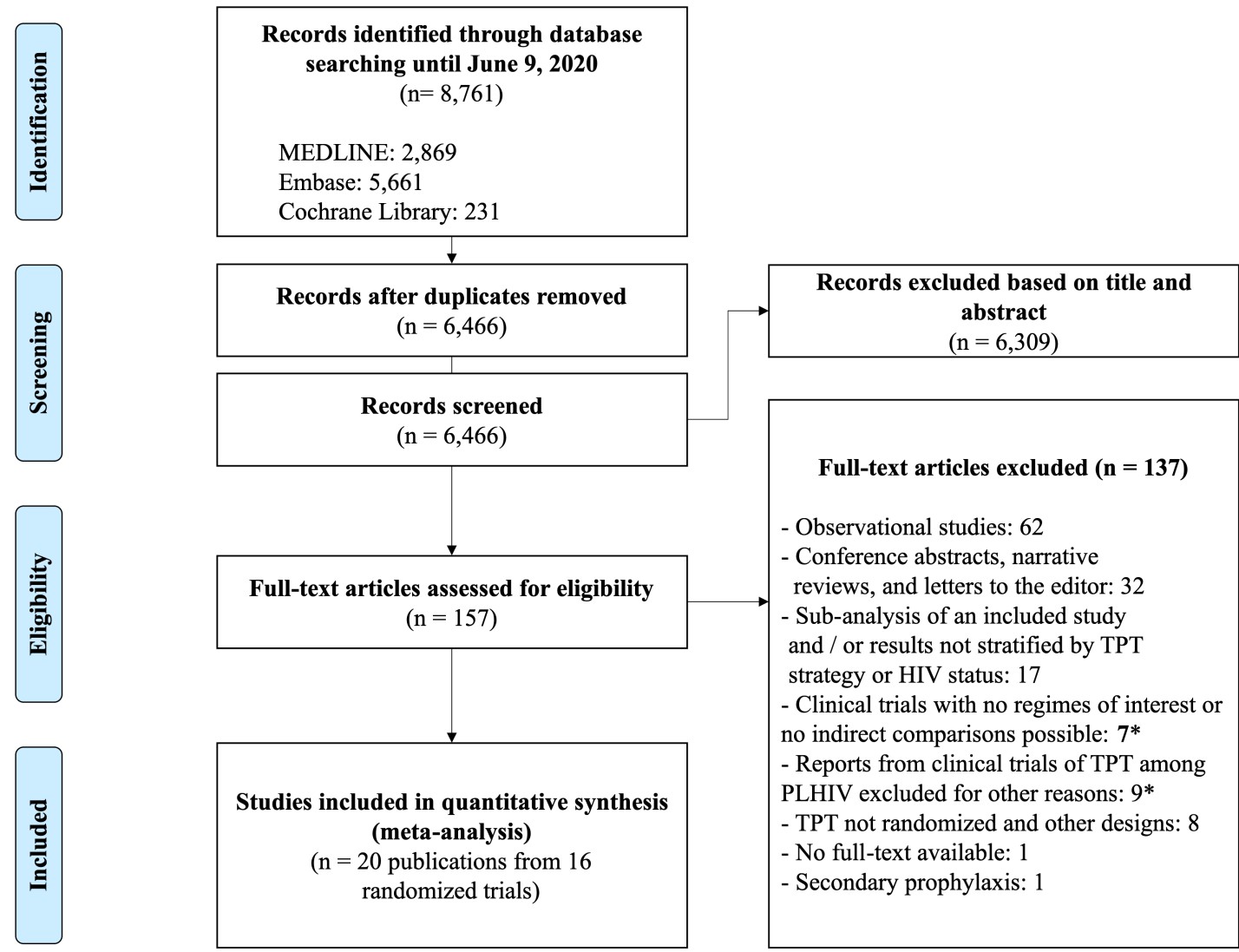

**Fig 1. Flowchart of study selection.** * For further detail on the reasons for exclusion, please refer to Table C in S1 File. PLHIV, people living with HIV; TPT, tuberculosis preventive therapy.

[51]; a third excluded study compared 12 weeks of an enhanced opportunistic infection prophylaxis versus trimethoprim-sulfamethoxazole, but both arms received isoniazid [52,53]; a fourth comparing 2 months of rifampin and pyrazinamide (2RZ) versus 9H used interchangeably rifampin or rifabutin and reported results together [54]; a fifth comparing 12H versus placebo offered isoniazid to the placebo arm during the trial [55]; and a sixth comparing 6H (daily or thrice a week) versus placebo and did not report stratified results for dosing schedules [56].

Nine trials were judged to be at high risk of bias due to problems in randomization and allocation concealment, blinding of outcome assessors, or missing data [8–10,28,29,31,32,34,40] (Fig A in S1 File).

Among the included studies, the median sample size was 616 (interquartile range [IQR], 317 to 1,892). Eight were carried out in Africa [8,11,27,30,34,36–38], 3 in Europe [9,10,29], 3 in the Americas[28,32,40], and 2 included sites in multiple continents [6,7] (Table 1).

Eight studies included daily 6H [8–11,28,30,34,36], 2 studies daily 9H[6,7], 4 studies daily 12H [29,32,37,40], 2 studies daily 24H [27,38], 1 study daily 36H[36], 1 study daily 72H [8], 4 studies daily 3HR [9–11,29], 2 studies weekly 3HP [7,8], 1 study daily 1HP [6], and 3 studies daily 2RZ [9,10,40]. Two studies were exclusively carried out in pediatric populations, with ages ranging from 3 months to 4 years [27,38]. Pediatric studies only reported on the development of TB and all-cause mortality. Among studies in adults, the mean or median age ranged from 29 to 36 years. Seven of 16 studies were published in the past 10 years [6–8,30,36–38]. Six studies either did not report or did not use ART [9–11,29,32,34]; in the other 10 trials, the proportion of participants who had ever received ART varied from 18.7% to 100%. The proportion of studies that reported any use of ART was the same for studies with an isoniazid arm (9/13) or a rifamycin-containing arm (4/6) (excluding pyrazinamide-containing regimens). The median proportion of ART use reported in studies with an isoniazid arm was 36.3% (range, 0% to 100%) and in studies with a rifamycin-containing regime was 25% (range, 0% to 96.3%).

As shown in Fig 2, with the exception of 36H, most arms of prolonged isoniazid (i.e., > = 24 months) had no posttreatment follow-up. On the other hand, all isoniazid regimens of 6 to 12 months (except for the 12H arm of Martínez Alfaro and colleagues), and all rifamycin-containing regimens had posttreatment follow-up duration of 12 months or more (rifamycin-containing: median, 18 months; 6 to 12H: median, 20.7 months).

## Network meta-analysis results

Fig B in S1 File shows the network of TPT regimens included for the outcome of microbiologically confirmed active TB. For this outcome [6–10,27,33,34,36–40], as shown in Table 2, when comparing grouped regimens, the regimens with the lowest rates of TB, compared to placebo or no treatment, were 24 to 72 H (IRR 0.5, 95% CI 0.3 to 0.8, $p$ = 0.01). There were no statistically significant differences between TPT regimens. Table D in S1 File shows the analysis by individual regimens.

Likewise, relative to no treatment or placebo, 24 to 72 H (IRR 0.5, 95% CI 0.3 to 0.8, $p$ = 0.005) were the regimens with the lowest rates of microbiologically confirmed or clinically diagnosed TB [6–11,27,29,30,32,34,36–40] (Table 3). However, both rifamycin-containing and 6 to 12H regimens showed evidence of a protective effect against microbiologically confirmed and clinically diagnosed TB.

Rifamycin-containing regimens had the lowest rates of all-cause mortality, significantly lower than with placebo or no treatment (IRR 0.6, 95% CI 0.5 to 0.8, $p < 0.001$) [6–11,27,29,31,32,34,36,37,39,40] (Table 4), or with 6 to 12H (IRR 0.6, 95% CI 0.5 to 0.8, $p < 0.001$) or with 24 to 72 H regimens (IRR 0.6, 95% CI 0.4 to 0.9, $p$ = 0.006). Findings were similar when comparing individual regimens (Table E in S1 File).

Rifamycin-containing regimens also had a statistically significant lower occurrence of grade 3 or worse hepatotoxicity compared to isoniazid regimens, with the largest difference observed between 24 and 72 H and rifamycin-containing regimens (RD per 100 persons 20.7, 95% CI 12.5 to 28.9, $p < 0.001$) [7,8,29,36,37,39,40] (Table 5). The highest risk of hepatotoxicity was with 72 H, as seen in individual regimen comparisons (Table F in S1 File).

Fig 3 shows the proportions of treatment completion by TPT regimens. Overall, shorter regimens had the highest proportions of treatment completion. Table G in S1 File shows the proportion of treatment completion by study arm.

When looking at the cumulative incidence of isoniazid resistance (including MDR), the incidence was similar among participants on any mono-isoniazid and rifamycin-containing regimens (0.37 cases per 100 people randomized, 95% CI 0.24 to 0.57 and 0.18 cases per 100 people randomized, 95% CI 0.05 to 0.69, respectively) [6–10,27,30,33,34,36–38,40] (Table 6). The cumulative incidence of rifampin resistance was also similar between any mono-isoniazid

**Table 1. Main characteristics of the included studies.**

| First author (year) | Country or region (TB incidenceª, year) | Number of participants included | TPT (months/ drugs) | Age (years) | CD4, cells/mm³ (median, IQR unless otherwise stated) | ART, n (%)[B] | LTBI[††] n (%) | Follow-up since randomization, (months)[b] | Primary outcome |
|---|---|---|---|---|---|---|---|---|---|
| Gordin (1997) | United States of America (6.7, 2000) | 517 | 6H, placebo | Mean 38 (range, 21–64) | 240 (100–417) | 377 (72.9)[¶] | 0 (0) TST | **Planned**: 30; **actual** (mean): 6H: 34 (NR); Placebo: mean 33 (NR) | Development of active TB, pulmonary, extrapulmonary, or both |
| Martínez Alfaro (2000) | Spain (23, 2000) | 133 | 12H, 3HR | Mean 32.2 (NR) | Mean 396 (NR) | 0 (0)[§] | 47 (35) TST | **Planned**: 24, **actual** (median): 12H: 19 (NR); 3HR: 16 (NR) | Development of active TB, in any site, in the first 2 years after treatment |
| Rivero (2007) | Spain (23, 2000) | 316 | 6H, 3HR, 2RZ[##] | Mean 31.3–33[†] (NR) | 466–522[†] (22–1,680)[†] | NR | 316 (100) TST | **Planned** (posttreatment): 24; **actual** posttreatment (mean)[#]: 6H: 12.8 (NR); 3HR: 12.6 (NR) | Development of active TB and treatment cessation due to toxicity |
| Rivero (2003) | Spain (23, 2000) | 319 | 6H, 3HR, 2RZ[##], no treatment | Mean, 32.7 (NR) | 230 (range, 0–1,495) | NR | 0 (0) TST | **Planned** (posttreatment): 24; **actual** posttreatment (mean)[#]: 6H: 12.7 (NR); 3HR: 14 (NR); no treatment: 19.6 (NR) | Development of active TB |
| Gordin (2000) and Gordin (2004) (for adverse events) | USA, Mexico, Brazil, and Haiti (25.4, 1990) | 1,583 | 12H, 2RZ[##] | Mean 37 (range, 16–70) | 436 (274–621) | 574 (36.3) | 1,128 (100) TST | **Planned**: 36 months; **actual**: 12H: 36.8 (NR) | Confirmed active TB |
| Sterling (2016) | USA, Brazil, Spain, Peru, Canada, and Hong Kong (35.3, 2007)* | 399 | 9H, 3HP | Median 36 (IQR, 29–44)[†] | 495–538[†] (389–729)[†] | 125 (31.3)[§] | 16 (4)[§§] | **Planned**: 33; **actual** (mean)[#]: 9H: 29.9 (NR); 3HP: 30.1 (NR)[#] | Adverse events |
| Temprano Group (2015) | Ivory Coast (159, 2015) | 2,056 | 6H, no treatment | Median 35 (IQR, 29–42)[†] | 459–467[†] (359–584)[†] | 1,630 (79.3)[§] | 337/967 (34.9) IGRA | **Planned**: 30; **actual** (median, IQR): 6H: 29.9 (29.9–30); no treatment: 29.9 (29.9–30) | Composite of death from any cause, AIDS-defining disease, non-AIDS–defining cancer, or non-AIDS–defining invasive bacterial disease |
| Badje (2017) (Temprano follow-up) | (159, 2015) | – | – | – | – | 1,831 (89.1)[§] | – | **Planned**: 60; **actual** (median, IQR): 6H: 58.8 (42–69.6); no treatment: 57.6 (39.6–69.6) | All-cause mortality |
| Swindells (2019) | 10 countries in Africa, Asia, and America** (249.9, 2015)* | 2,986 | 9H, 1HP | Median 35 (IQR, 28–43) | 470 (346–635) | 2,891/ 3,000 (96.3)[§] | 692 (23) IGRA or TST | **Planned**: 36 (after the last patient enrolled); **actual** (mean)[#]: 9H: 39.1 (NR); 1HP: 39.5 (NR) | First diagnosis of active TB, death from TB, or death from an unknown cause |
| Fitzgerald (2001) | Haiti (270, 2000) | 2,37 | 12H, placebo | Mean 32 (NR) | NR | 0 (0)[§] | 0 (0) TST | **Planned**: NI; **actual** (mean): 12H: 30 (NR); Placebo: 28.8 (NR) | Development of active TB, AIDS, and death |
| Whalen (1997) | Uganda (276, 2000) | 2,736 | 6H, 3HR, 3HRZ[BB], placebo | Mean 29–30[†] (NR) | NR | 0 (0)[¶] | 2,018 (74) TST | **Planned**: 36; **actual** (mean)[#]: placebo (TST +/−): 14.7 (NR); 6H (TST+/−): 13.1 (NR); 3HR: 14.9 (NR) | Development of active TB |

(*Continued*)

**Table 1.** (Continued)

| First author (year) | Country or region (TB incidence[a], year) | Number of participants included | TPT (months/ drugs) | Age (years) | CD4, cells/mm$^3$ (median, IQR unless otherwise stated) | ART, $n$ (%)[B] | LTBI[††] $n$ (%) | Follow-up since randomization, (months)[b] | Primary outcome |
|---|---|---|---|---|---|---|---|---|---|
| Johnson (2001) (Whalen follow-up) | (276, 2000) | – | – | – | NR | NR | – | **Planned**: NR; **actual** (mean): 6H[#] (TST+/−): 21.8; placebo[#] (TST +/−): 23.4; 3HR: 25.2 | Development of active TB, all-cause mortality, and drug resistance in patients who developed active TB during follow-up |
| Hawken (1997) | Kenya (451, 2000) | 684 | 6H, placebo | Mean 31.1 (SD, 6.9–7.5)[†] | 321.5–346[†](range, 3–3,352)[†] | NR | 136/595 (23) TST | **Planned**: 30; **actual** (median, range): 6H: 22 (0–41); Placebo: 21.8 (0–40.4) | Development of active TB |
| Samandari (2011) | Botswana (598, 2008) | 1,995 | 6H, 36H | Median 32 (IQR, 28–39) | 297 (172–449) | 946 (47.4)[§] | 468 (23) TST | **Planned**: 36; **actual** (mean)[#]: 6H: 32.6 (NR); 36H: 33 (NR) | Development of active TB |
| Samandari, (2015) (Samandari follow-up) | (598, 2008) | – | – | – | – | 1,007/ 1,398 (72)[§] | – | **Planned**: 72; **actual** (mean)[#]: 6H: 58.5 (NR); 36H: 58.6 (NR) | Development of TB or death from any cause |
| Zar (2006) | South Africa (883, 2004) | 263 | 24H (daily and t.i.w.), placebo | Median 24.7 months (IQR, 9.4–51.6) | 20 (% lymphocytes), (14–28) | 81 (30.8)[§] | 22/257 (9) TST | **Planned**: 24; **actual** (mean)[#]: 24H: 6.4 (NR)[#]; placebo: 5.1 (NR) | All-cause death |
| Rangaka (2014) | South Africa (948, 2010) | 1,369 | 12H, placebo | Median 34 (IQR, 30–40) | 216 (152–360) | 1,369 (100)[§¶] | 491/879 (55.9) IGRA or TST | **Planned**: 36; **actual** (mean)[#]:12H: 32.4 (NR)[#]; placebo: 32.2 (NR) | Development of active TB |
| Martinson (2011) | South Africa (963, 2006) | 1,148 | 6H, 3HR[BB], 3HP, 72H | Median 30.4 (IQR, 26.4–34.7) | 484 (350–672) | 215 (18.7)[§] | 1,148 (100) TST | **Planned**: 72; **actual** (median): 6H: 46.8 (NR); 3HP: 48 (NR); 72H: 46.8 (NR) | TB free survival |
| Madhi (2011) | South Africa and Botswana (977, 2007)[*] | 548 | 24H, placebo | Median 96 days (range, 91–120) | 28 (% lymphocytes), (6–58) | 542 (98.9)[§] | NR | **Planned**: 48; **actual** (median, range): 24H: 18 (0.25–27); placebo: 19 (0.25–27) | TB free survival |

[a] WHO TB burden estimates for all forms of TB (cases/100,000 people, year).

[b] Information about TPT regimens that was not of interest in our study and was not included.

[*] Weighted mean; in the study of Madhi and colleagues, only 4 PLHIV were included from Botswana; the incidence rate of South Africa is presented.

[**] Botswana, Brazil, Haiti, Kenya, Malawi, Peru, South Africa, Thailand, the USA, and Zimbabwe.

[†] Extreme values for all treatment arms.

[B] As reported by each study.

[¶] Baseline.

[§] Ever received.

[BB] Excluded since it was not a regimen of interest (intermittent dosing [Martinson] or using pyrazinamide [Whalen]) and did not allow indirect comparisons for regimens of interest.

[††] As diagnosed by TST or IGRA.

[§§] TST converter; the remaining were close contacts of TB cases.

[#] Person-years of follow-up divided by the number of subjects randomized in each arm.

[##] Used only for indirect comparisons.

ART, antiretroviral therapy; H, isoniazid; IGRA, interferon gamma release assay; IQR, interquartile range; LTBI, latent tuberculosis infection; mo, months; NR, not reported; P, rifapentine; PLHIV, people living with HIV; R, rifampin; RCT, randomized controlled trial; TB, tuberculosis; t.i.w., three times a week; TPT, tuberculosis preventive therapy; WHO, World Health Organization; Z, pyrazinamide.

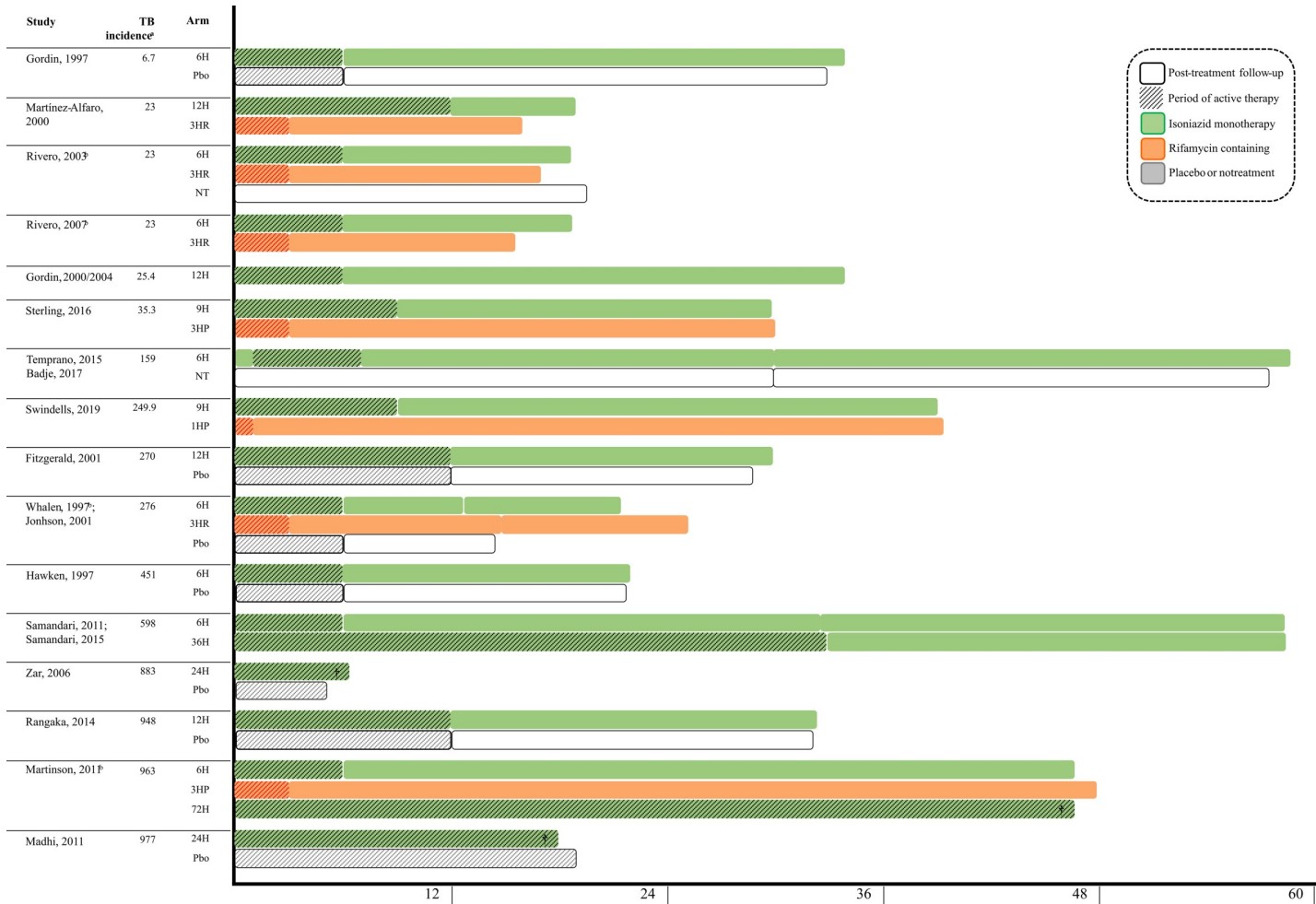

**Fig 2. Duration of treatment and posttreatment follow-up of the included studies.** Notes: Study arms are abbreviated by the number of months and treatment regimens (H, isoniazid; P, rifapentine; Pbo, placebo; R, rifampin; NT, no treatment). [a] WHO TB burden estimates for all forms of TB (cases/100,000 people, year). [b] Treatment arms considered not of interest were excluded in this figure for simplicity. [†] Mean or median follow-up of the planned continuous isoniazid regime. TB, tuberculosis; WHO, World Health Organization.

regimens and rifamycin-containing regimens (0.11 cases per 100 people randomized, 95% CI 0.06 to 0.22 and 0.24 cases per 100 people randomized, 95% CI 0.07 to 0.81, respectively). For participants randomized to any mono-isoniazid regimens, isoniazid resistance was statistically significantly higher than the development of rifampin resistance. Cumulative incidence of drug resistance by study arm can be found in Table H in S1 File.

Ranking of TPT regimens, by each outcome, is presented in Table 7.

The net heat plots for each outcome, by individual regimens, can be found in Figs C and D in S1 File. For the outcomes of microbiologically confirmed active TB and all-cause mortality, inconsistency appears to be low both within and between designs. However, for the outcome of grade 3 or worse hepatotoxicity, levels of inconsistency could not be evaluated due to low amount of information on between study heterogeneity.

## Secondary analysis

Results from the stratified analysis (by posttreatment follow-up time and TB incidence in the study setting) for the outcomes of microbiologically confirmed active TB and all-cause

**Table 2. NMA of incidence of microbiologically confirmed active TB throughout study duration by aggregated TPT regimens.**

| Comparison | Number of trials with direct comparison | Direct estimates IRR (95% CI) | Indirect estimates IRR (95% CI) | NMA IRR (95% CI) |
|---|---|---|---|---|
| Total number of comparisons (studies)* | 22 (13) | | | |
| 6 to 12 H versus placebo | 4 | 0.7 (0.5 to 1.1) | 1.1 (0.4 to 3.1) | 0.7 (0.5 to 1.1) |
| 24 to 72 H versus placebo | 2 | 0.7 (0.2 to 2.2) | 0.4 (0.2 to 0.8) | 0.5 (0.3 to 0.8) |
| 6 to 12 H versus 24 to 72 H | 2 | 1.6 (1 to 2.5) | 1.2 (0.4 to 3.5) | 1.5 (1.0 to 2.3) |
| 6 to 12 H versus rifamycin containing | 6 | 1.1 (0.8 to 1.5) | 0.5 (0.1 to 2.0) | 1.0 (0.8 to 1.4) |
| 24 to 72 H versus rifamycin containing | 1 | 0.5 (0.2 to 1.3) | 0.8 (0.4 to 1.4) | 0.7 (0.4 to 1.2) |
| Rifamycin containing versus placebo | 1 | 1 (0.2 to 4.4) | 0.7 (0.4 to 1.2) | 0.7 (0.4 to 1.2) |

* Includes studies with 2RZ. Placebo includes no treatment; rifamycin-containing regimens include 3HR, 3HP, and 1HP. The first regimen mentioned is compared to the second, i.e., first regimen over second regimen. Results should be interpreted as follows: The incidence of active TB with 24 to 72 H was 50% that of the incidence with placebo.

$I^2$ for the NMA 0% (0.0%; 34.7%).

CI, confidence interval; IRR, incidence rate ratio; NMA, network meta-analysis; TB, tuberculosis; TPT, tuberculosis preventive therapy.

mortality can be found in Tables I–L in S1 File. No significant differences were found between strata or the crude analysis, as confidence intervals overlap for direct, indirect, and NMA estimates. When looking at stratified analysis by posttreatment follow-up time, very few studies were included in the strata with no posttreatment follow-up for both the outcomes of microbiologically confirmed active TB and all-cause mortality; thus, confidence intervals are very wide. When stratifying by study setting TB incidence, no significant differences were found between strata. It is important to note that studies evaluating 24 to 72H were only carried out in settings with a TB incidence equal to or higher than 300 per 100,000 people. When studies were stratified on the basis of whether more, or less than 50% of participants had ever received ART, point estimates of the effect of aggregated regimens on mortality, and microbiologically confirmed TB were very similar in the 2 strata, although confidence intervals were wider, due to the smaller number of studies within each strata (Tables T and U in S1 File). When adjusted for proportion of study participants who had received ART, using meta-regression for NMA,

**Table 3. NMA of incidence of microbiologically confirmed and clinically diagnosed active TB throughout study duration by aggregated TPT regimens.**

| Comparison | Number of trials with direct comparison | Direct estimates IRR (95% CI) | Indirect estimates IRR (95% CI) | NMA IRR (95% CI) |
|---|---|---|---|---|
| Total number of comparisons (studies)* | 27 (16) | | | |
| 6 to 12H versus placebo | 7 | 0.6 (0.5 to 0.9) | 1 (0.5 to 2.2) | 0.7 (0.5 to 0.9) |
| 24 to 72H versus placebo | 2 | 0.8 (0.4 to 1.4) | 0.3 (0.2 to 0.6) | 0.5 (0.3 to 0.8) |
| 6 to 12H versus 24 to 72H | 2 | 1.9 (1.0 to 3.4) | 0.9 (0.4 to 1.7) | 1.3 (0.9 to 2.1) |
| 6 to 12H versus rifamycin containing | 7 | 1.1 (0.7 to 1.6) | 0.8 (0.2 to 2.7) | 1.0 (0.7 to 1.5) |
| 24 to 72H versus rifamycin containing | 1 | 0.4 (0.2 to 1.3) | 1 (0.5 to 1.8) | 0.8 (0.5 to 1.4) |
| Rifamycin containing versus placebo | 2 | 0.5 (0.2 to 1.1) | 0.8 (0.5 to 1.3) | 0.7 (0.4 to 1.0) |

* Includes studies with 2RZ. Placebo includes no treatment; rifamycin-containing regimens include 3HR, 3HP, and 1HP. The first regimen mentioned is compared to the second, i.e., first regimen over second regimen. Results should be interpreted as follows: The incidence of active TB with 24 to 72 H was 50% that of the incidence with placebo.

$I^2$ for the NMA 26.9% (0.0%; 58.7%).

CI, confidence interval; IRR, incidence rate ratio; NMA, network meta-analysis; TB, tuberculosis; TPT, tuberculosis preventive therapy.

**Table 4. NMA of incidence of all-cause mortality throughout study duration by aggregated TPT regimens.**

| Comparison | Number of trials with direct comparison | Direct estimates IRR (95% CI) | Indirect estimates IRR (95% CI) | NMA IRR (95% CI) |
|---|---|---|---|---|
| Total number of comparisons (studies)* | 27 (16) | | | |
| 6 to 12 H versus placebo | 7 | 1.0 (0.8 to 1.1) | 1.0 (0.6 to 1.6) | 1.0 (0.8 to 1.1) |
| 24 to 72 H versus placebo | 2 | 1.2 (0.7 to 2.0) | 0.9 (0.6 to 1.3) | 1.0 (0.7 to 1.4) |
| 6 to 12 H versus 24 to 72 H | 2 | 1.0 (0.7 to 1.5) | 0.8 (0.5 to 1.3) | 1.0 (0.7 to 1.3) |
| 6 to 12 H versus rifamycin containing | 7 | 1.6 (1.3 to 2.1) | 1.2 (0.5 to 2.7) | 1.6 (1.2 to 2.0) |
| 24 to 72 H versus rifamycin containing | 1 | 1.0 (0.4 to 2.3) | 1.8 (1.2 to 2.7) | 1.6 (1.2 to 2.4) |
| Rifamycin containing versus placebo | 2 | 0.6 (0.4 to 0.8) | 0.7 (0.5 to 1.1) | 0.6 (0.5 to 0.8) |

* Including 2RZ comparisons. Placebo includes no treatment; person-years of follow-up were used as denominator. Rifamycin-containing regimens include 3HR, 3HP, and 1HP. The first regimen mentioned is compared to the second, i.e., first regimen over second regimen. Results should be interpreted as follows: The incidence of all-cause mortality with rifamycin-containing regimens was 40% less than the incidence with placebo.

$I^2$ for the NMA 0% (0.0%; 48.8%).

CI, confidence interval; IRR, incidence rate ratio; NMA, network meta-analysis; TPT, tuberculosis preventive therapy.

the adjusted results for aggregated and individual regimens were very similar to the unadjusted results (Tables V–Y in S1 File).

Sensitivity analyses by individual TPT regimens (Tables M–P in S1 File) show similar results to the primary analyses. Thirty-six months of isoniazid was statistically significantly better at preventing microbiologically confirmed active TB when compared to 1HP, whereas 3HR was statistically significantly better at preventing all-cause mortality when compared to placebo or no treatment, 6H, and 36 H. For the outcome of all-cause mortality among rifamycin-containing regimens, when studies with high rates of ART were excluded [6], the protective effect of rifamycin-containing regimens remained unchanged (Table Q in S1 File).

**Table 5. NMA of RD of grade 3 or worse hepatotoxicity during treatment, by aggregated TPT regimens.**

| Comparison | Number of trials with direct comparison | Direct estimates RD per 100 persons (95% CI) | Indirect estimates RD per 100 persons (95% CI) | NMA RD per 100 persons (95% CI) |
|---|---|---|---|---|
| Total number of comparisons (studies)* | 9 (7) | | | |
| 6 to 12 H versus placebo | 2 | 1.6 (−5.9 to 9.2) | – | 1.6 (−5.9 to 9.2) |
| 24 to 72 H versus placebo | 0 | – | 13.5 (3.0 to 24.0) | 13.5 (3.0 to 24.0) |
| 6 to 12 H versus 24 to 72 H | 2 | −11.7 (−19.3 to −4.1) | −13.4 (−37.3 to 10.5) | −11.9 (−19.1 to −4.7) |
| 6 to 12 H versus rifamycin containing | 3 | 7.0 (0.8 to 13.2) | 47.3 (19.0 to 75.6) | 8.9 (2.8 to 14.9) |
| 24 to 72 H versus rifamycin containing | 1 | 26.5 (15.8 to 37.2) | 12.3 (−0.5 to 25.2) | 20.7 (12.5 to 28.9) |
| Rifamycin containing versus placebo | 0 | – | −7.2 (−16.9 to 2.5) | −7.2 (−16.9 to 2.5) |

* Includes studies with 2RZ. Placebo includes no treatment; rifamycin-containing regimens include 3HR and 3HP. The studies by Temprano group and colleagues and Swindells and colleagues were excluded from this analysis given that adverse events reported included events during treatment and posttreatment follow-up. The first regimen mentioned is compared to the second, i.e., risk with the first regimen minus risk with the second regimen. Results should be interpreted as follows: Participants who received 24 to 72 H had 20.5 more episodes of grade 3 or worse hepatotoxicity per 100 people randomized compared to participants receiving rifamycin-containing regimens.

$I^2$ for the NMA 20.5% (0%; 66.1%)

CI, confidence interval; NMA, network meta-analysis; RD, risk difference; TPT, tuberculosis preventive therapy

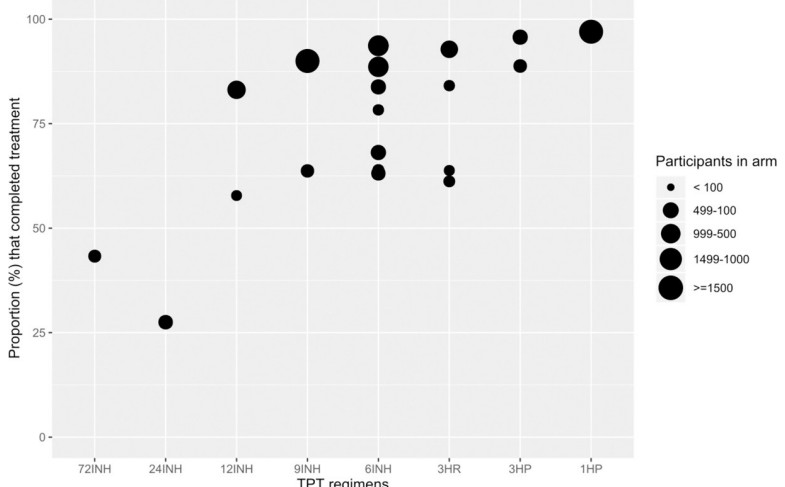

**Fig 3. Proportion of TPT completion by individual regimens.** Notes: No information was available for completion of 36H; information on placebo or no treatment is not shown as duration varied. TPT, tuberculosis preventive therapy.

In the stratified analysis by TST/IGRA status, the incidence of microbiologically confirmed or clinically diagnosed TB (Table R in S1 File) was statistically significantly lower among TST/IGRA positive PLHIV with 24 or 36H and with rifamycin-containing regimens when compared to placebo. However, for TST/IGRA negative or TST anergic PLHIV, there was no statistically significant difference in incidence with these TPT regimens compared to placebo.

**Table 6. Aggregate meta-analysis of the cumulative incidence of drug-resistant TB by aggregated TPT regimens.**

| Group | Number of studies | *n/N* | Pooled cumulative incidence of drug-resistant active TB per 100 people randomized (95% CI) | I² |
|---|---|---|---|---|
| **Proportion of any resistance** | | | | |
| 6 to 12 H | 11 | 27/6,441 | 0.42 (0.28 to 0.64) | 12% |
| 24 to 72 H | 3 | 4/1,444 | 0.28 (0.1 to 0.74) | 0% |
| Any mono-H | 14 | 31/7,885 | 0.39 (0.27 to 0.57) | 7% |
| Rifamycin containing | 7 | 9/3,092 | 0.31 (0.12 to 0.77) | 42% |
| Placebo or no treatment | 7 | 16/2,796 | 0.22 (0.03 to 1.81) | 87% |
| **Proportion of resistance to H (including MDR)** | | | | |
| 6 to 12 H | 11 | 24/6,441 | 0.39 (0.23 to 0.66) | 39% |
| 24 to 72 H | 3 | 4/1,444 | 0.28 (0.1 to 0.74) | 0% |
| Any mono-H | 14 | 28/7,885 | 0.37 (0.24 to 0.57) | 23% |
| Rifamycin containing | 7 | 6/3,092 | 0.18 (0.05 to 0.69) | 55% |
| Placebo or no treatment | 7 | 12/2,796 | 0.2 (0.03 to 1.53) | 84% |
| **Proportion of resistance to Rif (including MDR)** | | | | |
| 6 to 12 H | 11 | 6/6,441 | 0.09 (0.04 to 0.21) | 0% |
| 24 to 72 H | 3 | 3/1,444 | 0.21 (0.07 to 0.64) | 0% |
| Any mono-H | 14 | 9/7,885 | 0.11 (0.06 to 0.22) | 0% |
| Rifamycin containing | 7 | 7/3,092 | 0.24 (0.07 to 0.81) | 55% |
| Placebo or no treatment | 7 | 6/2,796 | 0.21 (0.1 to 0.48) | 0% |

Any mono-H includes 6 to 72 H; rifamycin-containing regimens include 3HR, 3HP, and 1HP; overall resistance includes all types of drug resistance reported in individual studies; not all studies carried out drug susceptibility testing for all first line anti-TB drugs. Results should be interpreted as follows: Among studies evaluating 6 to 12 H, there were 0.42 cases of drug-resistant TB per 100 people randomized.

CI, confidence interval; MDR, multidrug resistant, resistant to H and R; TB, tuberculosis; TPT, tuberculosis preventive therapy.

**Table 7. Ranking of regimens for each outcome evaluated.**

| Treatment | P-score |
|---|---|
| **Prevention of microbiologically confirmed TB** | |
| 24 to 72H | 0.84 |
| Rifamycin containing | 0.42 |
| 6 to 12H | 0.35 |
| Placebo | 0.05 |
| **Prevention of microbiologically and clinically diagnosed TB** | - |
| 24 to 72H | 0.83 |
| Rifamycin containing | 0.52 |
| 6 to 12H | 0.45 |
| Placebo | 0.03 |
| **Prevention of all-cause mortality** | |
| Rifamycin containing | 1 |
| 6 to 12H | 0.37 |
| 24 to 72H | 0.25 |
| Placebo | 0.21 |
| **Lowest risk of grade 3 or higher hepatotoxicity** | |
| Rifamycin containing | 0.96 |
| Placebo | 0.59 |
| 6 to 12H | 0.47 |
| 24 to 72H | 0.01 |

TB, tuberculosis.

## Discussion

In this NMA of randomized trials of TPT in PLHIV, compared to mono-isoniazid regimens, rifamycin-containing regimens proved to be similarly effective in preventing microbiologically confirmed TB, statistically significantly more effective in preventing all-cause mortality, with lower rates of hepatotoxicity, without evidence of generating resistance to rifampin, and they were associated with higher completion rates when compared to longer regimens.

When initiating TPT in PLHIV, selection of the optimal regimen should be based on benefits, in terms of prevention of TB and mortality, potential harms including hepatotoxicity and development of drug resistance, and acceptability as indicated by treatment completion. This systematic review and NMA provides information on all these important outcomes of TPT in PLHIV. Additionally, by using an NMA approach, we were able to compare all TPT regimens currently recommended.

A major limitation to the interpretation of our results is the impact of potential confounding due to differences in posttreatment follow-up time and TB incidence in the study setting on the estimates of incidence of TB or all-cause mortality. A stratified analysis undertaken to address this confounding was limited by insufficient power within strata. An individual patient data meta-analysis may improve the control of bias created by these potential confounders.

Other limitations of the review include insufficient studies evaluating TPT regimens in pregnant women and children, thus limiting our findings' generalizability to these 2 groups. Current guidelines recommend isoniazid-based regimens as well as 3HR for TPT in children of all ages living with HIV, while more information is still needed for 3HP in children under 2 years of age and for 1HP in children under 13 years of age [5,12]. Trials evaluating the use of

rifamycin-based regimens in pregnant women with HIV are urgently needed given recent evidence of substantial toxicity of isoniazid during pregnancy in women with HIV [51].

Results from the NMA, indicating that rifamycin-containing regimens are better at preventing all-cause mortality compared to other regimens, are driven by the evidence from direct estimates. For example, in the study by Swindells and colleagues, there were 6 deaths in the 1HP group and 10 in the 9-month H group, with an incidence rate difference of –0.08. However, this was not statistically significant (log-rank test *p*-value: 0.31).

Although 96.3% of the study population had ever received ART in the study by Swindells and colleagues, in all other studies evaluating rifamycin-containing regimens, use of ART was either not reported or ranged from 0 to 31.3%. In a post hoc analysis excluding the study by Swindells and colleagues, rifamycin-containing regimens still showed lower rates of all-cause mortality. This suggests that the decrease in mortality shown by rifamycin-containing regimens is unlikely to be confounded by the use of ART. There is not an obvious reason as to why rifamycin-containing regimens reduce all-cause mortality, but not TB incidence, when compared to isoniazid. An individual patient meta-analysis evaluating daily INH with ART to ART alone found that the combined treatments did not have a statistically significant impact in reducing all-cause mortality when compared to ART alone [57]. However, rifamycin is a sterilizing drug, which may explain a bigger effect on preventing TB deaths. Unfortunately, most studies reported all-cause deaths; thus, we were not able to stratify results into TB-related or unrelated deaths.

In this analysis, there appears to be no difference between different grouped regimens in preventing microbiologically confirmed active TB or microbiologically and clinically diagnosed TB. A previous aggregate data meta-analysis comparing at least 36 months of H (prolonged H) to 6 H found that prolonged regimens were more effective in preventing TB [58]. However, in the studies that evaluated these prolonged isoniazid regimens, there was no post-treatment follow-up or this follow-up was for less than a year; thus, TB cases detected are only those that developed while on treatment or shortly after. Yet in our analysis, all studies evaluating rifamycin-containing regimens and almost all of 6 to 12H had posttreatment follow-up of more than a year. Another potential source for confounding was TB incidence in the study setting, as risk of reinfection is considerable in high TB incidence settings and has been cited as the main reason for recommending prolonged H regimens [12]. In the studies by Johnson and colleagues, Golub and colleagues, and Swindells and colleagues, shorter isoniazid and rifamycin-containing regimens have shown long-lasting protection against TB [6,33,59]. The first 2 studies were carried out in TB incidence settings of less than 300 per 100,000 people, Uganda (TB incidence 276 cases per 100,000 people at the time of the study)[60] and Brazil (Rio de Janeiro TB incidence 79.2 cases per 100,000 people as reported by the study), where protection offered through TPT appears to last from 3 to 7 years [33]. In the study by Swindells and colleagues, both 9 H and 1HP provided protection against TB for up to 5 years, in low to high TB incidence settings, perhaps reflecting the protection offered by high coverage with ART in that study population. Given that studies evaluating prolonged H regimens were carried out in high or very high TB incidence settings (Botswana TB incidence 598 per 100,000 people at the time of the study; South Africa TB incidence 963 per 100,000 people at the time of the study; and India TB incidence 285 per 100,000 people at the time of the study)[60] as well as having the shortest posttreatment follow-up time, the effects of these regimens on preventing microbiologically confirmed active TB are likely confounded. For shorter regimens with longer follow-up, the efficacy of TPT is possibly reduced by reinfection, whereas prolonged regimens were only evaluated for the duration of treatment. In our analysis, we addressed this issue by carrying out a stratified analysis, although as previously mentioned, we were unable to detect differences between strata given the lack of power.

In our analysis, incidence of all forms of TB was significantly lower among PLHIV with a positive TST/IGRA test who received prolonged H regimens or rifamycin-containing regimens when compared to placebo. This benefit was not seen among TST/IGRA negative or anergic individuals. Although inferences are limited as this analysis included only 6 trials, this finding is in accordance with a previous meta-analysis of treatment of LTBI among PLHIV [3] and reflects a higher risk of reactivation of TB among PLHIV with evidence of LTBI [61]. This finding reinforces the message that TST or IGRA are potentially useful to identify PLHIV at greater risk of disease [61], who will derive greater benefit from TPT while also identifying those with negative tests who will have lower likelihood of benefit, yet run the same risk of adverse events from TPT [51,62]. Current guidelines encourage testing for LTBI, while also stipulating that this should not become a barrier, hence is not an absolute requirement for initiating TPT among PLHIV [4].

Currently, WHO considers rifamycin-containing regimens (3HR and 3HP) as equally preferred as 6H or 9H, regardless of patients' HIV status [12]. According to our results, rifamycin-containing regimens are as efficacious as 6 to 12H in preventing active TB, may be more effective in preventing death, and have a lower risk of liver toxicity. However, an important potential limitation of rifamycin-containing regimens among PLHIV is the risk of drug–drug interactions [12], particularly with ART. For example, a recent study in healthy volunteers using a combination of dolutegravir and rifapentine was stopped because of high rates of adverse drug reactions [63]. However, another recent study among PLHIV found that 3HP could be safely administered along with dolutegravir [64]. The combination of a rifamycin and efavirenz does not require dose adjustments [65]. Since the pharmacokinetics effects of anti-TB therapy on ART (and vice versa) are highly variable (often requiring dose adjustments), further consultation of updated resources is advised when prescribing rifamycin-containing regimens in PLHIV [65,66]. Drug costs for rifamycin-containing regimens may be greater than for isoniazid, but one study found that total health system costs (including personnel, lab costs, etc.) were lower with 4R than 6H or 9H in low-, middle- and high-income settings [67].

Regarding treatment rankings, it has been shown that treatment designs with higher uncertainty levels (wide confidence intervals) around the effect estimate are more likely to rank higher than designs with lower levels of uncertainty [68,69]. Given the differences between studies in size, risk of bias, use of ART at baseline, posttreatment follow-up, TB incidence in the study setting, and HIV standards of care, we considered that treatment rankings should be interpreted with caution. Other factors such as treatment costs and availability should be taken into consideration when choosing a treatment regimen.

Further trials with head-to-head comparisons of different rifamycin-containing short TPT regimens are needed to establish the optimal regimen for high TB incidence settings, for pregnant women and for children living with HIV. Information on treatment acceptability, tolerability, completion, and costs from these trials will help determine which rifamycin regimens should be selected for which populations and settings.

## Conclusions

This systematic review and NMA provides evidence that rifamycin-containing regimens are as effective as mono-H regimens in preventing TB and statistically significantly better at preventing all-cause mortality, with significantly lower hepatotoxicity rates, and higher completion rates, without generating RIF resistance. The findings of this review support the recommendations by WHO and the CDC that rifamycin-containing regimens are acceptable for TPT in PLHIV.

## Supporting information

**S1 Data. Holds the information extracted from the original studies and used in our NMA.**
NMA, network meta-analysis.
(XLSX)

**S1 File. Contains details of study methods (PRISMA checklist, Material A, and Tables A and B), a list of the excluded studies with reasons (Table C), and additional results (Tables D–Y and Figs A–D).** PRISMA: Checklist of items to include when reporting a systematic review involving an NMA. Material A: Search strategy. Table A: Information extracted in a predefined extraction sheet. Table B: Risk of bias assessment tool adapted from the Revised Cochrane risk-of-bias tool for randomized trials (RoB 2). Table C: Studies excluded after full-text review. Fig A: Risk of bias assessment of the included studies. Fig B: Network graph of TPT regimens for the outcome of incidence of microbiologically confirmed active TB. Table D: NMA of incidence of microbiologically confirmed active TB throughout study duration, by individual TPT regimens. Table E: NMA of incidence of all-cause mortality throughout study duration, by individual TPT regimens. Table F: NMA of risk of grade 3 or worse hepatotoxicity during treatment, by individual TPT regimens. Table G: Treatment completion of each study arm among all the included studies. Table H: Cumulative incidence of drug-resistant TB by treatment arm, among all the included studies. Fig C: Net heat plot of the NMA of incidence of microbiologically confirmed active TB, by individual TPT regimens. Fig D: Net heat plot of the NMA of incidence of all-cause mortality, by individual TPT regimens. Table I: Effect of aggregated TPT regimens on incidence of microbiologically confirmed active TB, stratified by length of posttreatment follow-up. Table J: Effect of aggregated TPT regimens on incidence of all-cause mortality, stratified by posttreatment follow-up time. Table K: Effect of aggregated TPT regimens on incidence of microbiologically confirmed active TB, stratified by study setting TB incidence. Table L: Effect of aggregated TPT regimens on incidence of all-cause mortality, stratified by study setting TB incidence. Table M: Effect of TPT regimens on incidence of microbiologically confirmed active TB, excluding studies with no follow-up or follow-up less than 1 year. Table N: Effect of TPT regimens on incidence of microbiologically confirmed active TB, excluding studies with a study setting TB incidence of more than 300 per 100,000. Table O: Effect of TPT regimens on incidence of all-cause mortality, excluding studies with no posttreatment follow-up or follow-up less than 1 year. Table P: Effect of TPT regimens on incidence of all-cause mortality, excluding studies with a study setting TB incidence of more than 300 per 100,000. Table Q: Effect of TPT regimens on incidence of all-cause mortality, excluding rifamycin-containing studies with a high rate of ART use. Table R: Effect of TPT regimens on incidence of microbiologically confirmed and clinically diagnosed TB, stratified by TST/IGRA status. Table S: Detailed information of completion rates, completion criteria and methods used to assess adherence to the TPT regimes of interest in the included studies. Table T: Effect of aggregated TPT regimens on incidence of all-cause mortality, stratified by the proportion of study participants receiving ART (above or below 50%). Table U: Effect of aggregated TPT regimens on incidence of microbiologically confirmed TB, stratified by the proportion of study participants receiving ART (above or below 50%). Table V: NMA of incidence of all-cause mortality throughout study duration, by aggregated TPT regimens and adjusted for the proportion of subjects receiving ART. Table W: NMA of incidence of microbiologically confirmed active TB throughout study duration, by aggregated TPT regimens and adjusted for the proportion of subjects receiving ART. Table X: NMA of all-cause mortality by individual TPT regimens adjusted for ever use of ART. Table Y: NMA of microbiologically confirmed TB by individual TPT regimens, adjusted for ever use of ART. ART, antiretroviral therapy; IGRA, interferon gamma release assay; NMA, network meta-analysis; PRISMA,

Preferred Reporting Items for Systematic Reviews and Meta-Analyses; TB, tuberculosis; TPT, tuberculosis preventive therapy; TST, tuberculin skin test.
(DOCX)

## Author Contributions

**Conceptualization:** Mercedes Yanes-Lane, Edgar Ortiz-Brizuela, Jonathon R. Campbell, Olivia Oxlade, Dick Menzies.

**Data curation:** Mercedes Yanes-Lane, Edgar Ortiz-Brizuela.

**Formal analysis:** Mercedes Yanes-Lane, Jonathon R. Campbell, Andrea Benedetti, Dick Menzies.

**Funding acquisition:** Olivia Oxlade, Dick Menzies.

**Investigation:** Mercedes Yanes-Lane, Edgar Ortiz-Brizuela, Jonathon R. Campbell, Andrea Benedetti, Gavin Churchyard, Olivia Oxlade, Dick Menzies.

**Methodology:** Mercedes Yanes-Lane, Edgar Ortiz-Brizuela, Jonathon R. Campbell, Gavin Churchyard, Olivia Oxlade, Dick Menzies.

**Project administration:** Olivia Oxlade.

**Resources:** Olivia Oxlade.

**Software:** Mercedes Yanes-Lane, Andrea Benedetti.

**Supervision:** Jonathon R. Campbell, Gavin Churchyard, Olivia Oxlade, Dick Menzies.

**Validation:** Mercedes Yanes-Lane, Edgar Ortiz-Brizuela, Jonathon R. Campbell, Andrea Benedetti, Gavin Churchyard, Olivia Oxlade, Dick Menzies.

**Visualization:** Mercedes Yanes-Lane, Edgar Ortiz-Brizuela.

**Writing – original draft:** Mercedes Yanes-Lane, Edgar Ortiz-Brizuela.

**Writing – review & editing:** Mercedes Yanes-Lane, Edgar Ortiz-Brizuela, Jonathon R. Campbell, Andrea Benedetti, Gavin Churchyard, Olivia Oxlade, Dick Menzies.

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
