## [Editor Report · Decision Letter 0]

6 Jan 2021

Dear Dr Menzies, 

Thank you for submitting your manuscript entitled "Tuberculosis Preventive Therapy for People Living with HIV: A Systematic Review and Network Meta-analysis" for consideration by PLOS Medicine.

Your manuscript has now been evaluated by the PLOS Medicine editorial staff and I am writing to let you know that we would like to send your submission out for external peer review.

Kind regards,

Thomas J McBride, PhD

Senior Editor

PLOS Medicine

---

## [Decision Letter · Decision Letter 1]

17 Feb 2021

Dear Dr. Menzies,

Thank you very much for submitting your manuscript "Tuberculosis Preventive Therapy for People Living with HIV: A Systematic Review and Network Meta-analysis" (PMEDICINE-D-20-06010R1) for consideration at PLOS Medicine. 

[LINK]

In light of these reviews, I am afraid that we will not be able to accept the manuscript for publication in the journal in its current form, but we would like to consider a revised version that addresses the reviewers' and editors' comments. Obviously we cannot make any decision about publication until we have seen the revised manuscript and your response, and we plan to seek re-review by one or more of the reviewers. 

We expect to receive your revised manuscript by Mar 10 2021 11:59PM. Please email us (plosmedicine@plos.org) if you have any questions or concerns.

We look forward to receiving your revised manuscript. 

Sincerely,

Dr Raffaella Bosurgi, 

Executive Editor

Medicine 

PLOS Medicine

plosmedicine.org

Comments from the reviewers:

Reviewer #1: Thank you for the opportunity to review this paper. I note that I am not a statistician and I recommend that a statistical reviewer also looks at this piece. This manuscript presents a further network meta-analysis (NMA) of difference treatments for latent tuberculosis infection (LTBI), this time focused on people living with HIV (PLHIV). The authors' emphasis on the value of rifamycin-containing regimens is overly strongly expressed at various points in the piece, considering the weight of the evidence presented and the fact that much of it is from rifampin-containing regimens, which are contraindicated by drug-drug interactions in PLHIV. Sadly, rifapentine-containing regimens, which the authors drift towards in the discussion, are out of reach in terms of cost in high-HIV and TB burden settings. As the World Health Organization's (WHO)'s recommendation is that 6 months of isoniazid is the standard of care for such individuals, the author's current grouping of 6-12 month isoniazid regimens together limits the use of their work for revising these guidelines. Indeed, WHO recommends that this six month regimen is used for both PLHIV and those who are HIV negative; in this context the author's emphasis on the use of rifamycin-containing regimens specifically among people for whom drug-drug interactions are of concern reads oddly without further discussion.

MAJOR

ABSTRACT

1) Consistency of baseline- it would aid the reader if all comparisons were made to one baseline regimen (see comment on baselines later), such that a better feel for the results can be obtained. At the moment, it is very hard to see how the results feed into the sentence 'Rifamycin-containing regimens appear safer and at least as effective as isoniazid regimens in preventing TB and death and should be considered part of routine care in PLHIV' due to this inconsistency.

INTRODUCTION

1) Line 40- you state that WHO 'considers six months of isoniazid as the preferred TPT regimen for PLHIV'; it is worth noting here that WHO does not differentiate their recommendation by HIV status. (Additionally, please record a) WHO's recommendation of 36 months of isoniazid preventative therapy (IPT) among PLHIV in high incidence and transmission settings for TB (which ties into your sentence on line 42) and b) WHO's statement that 'Rifampin- and rifapentine-containing regimens should be prescribed with caution to people living with HIV who are on ART because of potential drug-drug interactions'.)

2) One of the selling points that the authors state in the cover letter for this piece is that 'the network meta-analysis is also novel, as prior reviews have conducted more traditional meta-analyses'. In fact, several NMAs have been conducted of treatment regimens for LTBI (they are a very established technique) e.g. Pease BMC ID 2017, Zenner Ann Intern Med 2017, Stagg Ann Intern Med 2014. The authors should cite these papers in the introduction of their paper, as they provide methodological precedence for their work as they narrow down the population of interest to people living with HIV.

3) Given the two points above, the others also need to clearly define the need for a new NMA specifically among HIV positive individuals.

METHODS

4) Line 72- Please clearly define your population of interest- were studies in children included? How did you set criteria around whether participants had definite LTBI?

5) Line 78- 'Studies were excluded if there was no direct comparison of interest or if the only comparison included regimens using pyrazinamide and/or ethambutol as they are no longer recommended for TPT.' Regimens that are no longer of interest can provide important information in sparse data networks within NMAs, even if you would not consider the regimen itself as a treatment. Please justify the exclusion of these studies in this light.

6) Line 86- you state that you adapted the Rob2 tool- please say how and why.

7) Line 93- 'A study was classified as low risk of bias when there was low risk in the randomization process and low risk in at least two of the four remaining domains. If these conditions were not met, a study was considered at high risk of bias.' Please state if this is the normal approach for Rob2 and, if not, why you chose this approach.

8) Line 118- was the decision to run a random effects analysis made a priori? What was the decision-making behind this?

9) Line 118- as this is a medical journal, please explain the graph-theoretical approach to NMAs (and justify this choice of a frequentist approach, as opposed to a Bayesian one) and net heat plot method.

RESULTS

10) 5 full texts not available is quite substantial given that you only included 15 trials. What size were these studies and what regimens did they examine (are they in table S3?)

11) Figure 2- this figure was too fuzzy to read within the submission.

12) Table 1- 5 out of your 15 studies had either no-one confirmed as LTBI positive or less than 10%. A further 5 had less than 50%. This is a large concern when it comes to your findings.

13) S2 Fig- how does this map to the groupings under my query about line 145?

14) Given that the WHO recommends 6 months of isoniazid at the standard of care treatment and the sizeable number of trials using this regimen, I would strongly recommend making this the baseline throughout your analyses, rather than a mixed grouping.

15) Table 2- Your estimate for the efficacy of 6 to 12 H at preventing active TB vs Placebo is in itself quite uncertain- 0.8 (0.5 to 1.1). How do you explain this?

16) Table 3- can you hypothesize why none of the isoniazid regimens are effective against all-cause mortality and the implications of this for interpreting your isoniazid result? How good is rifamycin at preventing death in comparison to 6 month isoniazid regimens in studies where ART use was high (i.e. the reverse analysis of S17 table- see comment regarding line 357)? In terms of the rifamycin vs. 6-12m isoniazid result, what does this relative rate translate to in terms of absolute rates?

17) You chose not to calculate ranks within your analyses, but as a matter of interest how did rifamycin-containing regimens rank throughout?

DISCUSSION

18) Line 332- 'higher completion rates'- this is not substantiated by the data. Shorter regimens have higher completion rates, but not specifically rifamycin-containing ones.

19) Line 357- pleas state Swindells' result regarding the risk of death for isoniazid and rifampin-containing regimens.

20) Line 404- greater depth as to the fact that WHO does not recommend rifamycin-containing regimens in people living in HIV is required here. Please start by describing this and state the context of these concerns for rifampin, noting that most of your data is from rifampin-containing regimens. Given the frequency of use of these affected ART regimens globally, the lack of ability in many settings where TB-HIV is particularly common to alter the dose of ART, and the cost of rifapentine vs. rifampin please caveat the real-world applicability of your findings.

21) Line 416- 'Given this, rifamycin-containing regimens should be considered the regimens of choice for TPT in PLHIV.' This statement is very strong in light of how much of your data comes from rifampin trials, the uncertainty around the impact of rifapentine containing regimens on mortality from table S5, and the concerns about rifampin and ARTs. Please alter.

SUPPLEMENTS

22) Supplementary Material A- why did you search for 'LTBI therapy' etc. but not spell out latent tuberculosis (latent tuberculosis infection, latent TB, latent TB infection)? This could have been a source of missing studies. It would been easier to have used rows 10 to 18 as treatment terms only without need to specify LTBI in here (given that you did so as an independent item in rows 2 and 4).

23) Supplementary Material A- given that all trials of LTBI therapy will have examined at least one of your outcomes of interest, why did you include rows 20-25 of your search? As these terms are not always explicit in titles, abstracts, and key words, this could have led to missing studies.

MINOR

ABSTRACT

24) The authors repeatedly use the word 'significantly' throughout the text. Use of this word should be made cautiously given potential confusion between the colloquial and the statistical. In the latter case, use of p-value thresholds alone to mark out important results for causal analyses has largely been debunked; relative effect sizes, confidence intervals and sample size should all be interpreted together.

METHODS

25) Cochrane Database of Systematic Reviews- why did you choose to search a database of reviews, as opposed to a database of trials?

26) The authors should check that their manuscript lines up against the PRISMA guidelines for NMAs, rather than the standard PRISMA guidelines.

27) Line 145- 'mono-H (6 to 72 months of H), 6 to 12 months of H, 24 to 72 months of H, and rifamycin-containing regimens (3HR, 3HP, and 1HP).' This sentence is unclear as the first grouping overlaps with the subsequent two.

28) Line 161- stRAtified.

RESULTS

29) Line 318- were, not where.

SUPPLEMENTS

30) Supplementary Material A- please number the rows of your search so the AND and OR rows make sense.

31) S2 Table- please define N (I assume this is no) and PN.

Reviewer #2: See attachment

Michael Dewey

Reviewer #3: Thank you for the opportunity to review this important piece of work.

a) Line 51. Are you comparing phase III clinical trial efficacy or effectiveness (real conditions)? What degree of "pragmatism" do they have? In line 58 you say you exclude observational studies.

b) Line 63. Why was clinical TB initially excluded? Should it be specified upfront in the protocol as primary and secondary endpoints (lab confirmed and treatment initiation, respectively). Could you still do that? Could you include table S18 within the main manuscript?

c) Line 105. Do you think all-cause mortality an important outcome? Do you think that differences in mortality, when found, could be attributed to the regimen used? Could you expand the discussion a little bit on line 353 (I wonder why there is no difference in efficacy when using ryfamicin-based regimens, but there is on mortality).

d) Line 111. It would be nice to have a table with all the different measures of adherence used in the studies (supp material is fine).

e) How would the frequency of visits, or screening for TB to all participants might have changed the efficacy of the different regimens? How did you account for different PT duration follow up?

f) Line 161, Typo "stratified".

g) Table 1. Could you include the definition of the primary efficacy endpoint of each trial?

h) Table 2. Could you include the references of studies included for each comparison (all tables)? For studies including composite endpoints, like Swindells' 1HP, did you exclude the clinically diagnosed participants?

i) Table 3. Here I understand you are you including all studies independently of the ART status/CD4 count status. Related comment: Line 357. I understand that % or ART among arms is similar if only RCT were included. Your explanation on its role as confounder should contradict an even distribution of participants on ART. Am I interpreting correctly?

j) No comments in the discussion about the efficacy of PTs when outcomes include clinical TB diagnosis (stemming from table S18)

k) Fig 2's quality is too low to be interpreted.

[LINK]

---

## [Decision Letter · Decision Letter 2]

9 Jun 2021

Dear Dr. Menzies,

Thank you very much for re-submitting your manuscript "Tuberculosis Preventive Therapy for People Living with HIV: A Systematic Review and Network Meta-analysis" (PMEDICINE-D-20-06010R2) for consideration at PLOS Medicine.

I have discussed the paper with editorial colleagues and our academic editor, and it was also seen again by two reviewers. I am pleased to tell you that, provided the remaining editorial and production issues are fully dealt with, we expect to be able to accept the paper for publication in the journal.

[LINK]

Please let me know if you have any questions, and we look forward to receiving the revised manuscript.   

Sincerely,

Richard Turner, PhD

rturner@plos.org

Requests from Editors:

Please submit your revision as a research article.

Please ensure that you comply with PLOS' data policy (https://journals.plos.org/plosmedicine/s/data-availability) in your revision. 

Please remove the information on funding from the title page. In the event of publication this information will appear in the article metadata, via entries in the submission form. 

In the abstract and throughout the text, please quote p values alongside 95% CI, where available.

Please add a new final sentence to the "Methods and findings" subsection of your abstract, beginning "Study limitations include ..." or similar and quoting 2-3 of the study's main limitations. 

After the abstract, please add a new and accessible "author summary" in non-identical prose. You may find it helpful to consult one or two recent research papers published in PLOS Medicine to get a sense of the preferred style. 

Please remove the "Role of the funding source" section, noting the comment above.

Please use the style "... 12 clinical trials" throughout the text, although numbers should be spelt out at the start of sentences. 

Throughout the text, please restyle the reference call-outs as follows: "... darunavir) [12,13]." (i.e., preceding punctuation and with no spaces within the square brackets). 

Noting reference 3 and others, please ensure that competing interest information is removed from all citations. 

Is reference 4 lacking a report number?

Are references 19 and 50 lacking journal names? Reference 53 may also need attention.

Please reformat the author lists for references 40 and 42.

Reference 63 may need additional access information.

Please abbreviate journal names consistently in the reference list, and use the abbreviation "PLoS ONE".

Please rename the PRISMA attachment "S1_PRISMA_Checklist" or similar, and refer to it by this label in the Methods section. 

In the checklist, please refer to individual items by section (e.g., "Methods") and paragraph numbers, not line or page numbers as these generally change in the event of publication. 

Comments from Reviewers:

*** Reviewer #2: 

The authors have addressed all my points

Michael Dewey

*** Reviewer #3: 

I think all comments have been addressed. I am satisfied with the changes implemented /answers. Congratulations for this important piece of work.

***

[LINK]

---

## [Editor Report · Decision Letter 3]

19 Jun 2021

Dear Dr. Menzies,

Thank you very much for re-submitting your manuscript "Tuberculosis Preventive Therapy for People Living with HIV: A Systematic Review and Network Meta-analysis" (PMEDICINE-D-20-06010R3) for consideration at PLOS Medicine.

I have discussed the paper with editorial colleagues and our academic editor, and we will need to ask you to address some additional points before we are able to proceed further. 

[LINK]

We hope to receive your revised manuscript within 2 weeks. Please email us (plosmedicine@plos.org) if you have any questions or concerns.

Please let me know if you have any questions, and we look forward to receiving the revised manuscript. 

Sincerely,

Richard Turner, PhD

rturner@plos.org

*** Requests from Editors:

Please add a sentence, say, to the abstract to summarize study characteristics, e.g., median sample sizes and regions where the studies were done.

Throughout the text, please remove spaces from the reference call-outs (e.g., "... PLHIV [63,64]."). 

We suggest using the spelling "graphically" (Methods). 

At line 132, should that be "... did not compare 2 or more ..."?

Please remove trademarks, e.g., at line 163.

Please spell out the author names for reference 38, and any other relevant references. 

Please use the journal name abbreviation "PLoS ONE".

*** Comments from Academic editor:

It has come to our attention that a published systematic review and meta-analysis focused on IPT for PLHIV on ART (Ross et al, Lancet ID https://www.thelancet.com/journals/lanhiv/article/PIIS2352-3018(20)30299-X/fulltext). We did not see this cited in your article.

Your NMA of different regimens is welcome and a contribution to the literature, but we wonder whether there is effect modification due to ART status and suspect many of our readers will wonder the same given that the included data features cohorts not receiving ART from the pre-ART era, which is no longer the clinical norm. We therefore ask that you investigate the prospect of effect modification due to ART formally, presenting those findings in the MS, and if appropriate presenting separate effect estimates based on ART status.

***

---

## [Decision Letter · Decision Letter 4]

13 Jul 2021

Dear Dr. Menzies,

Thank you very much for re-submitting your manuscript "Tuberculosis Preventive Therapy for People Living with HIV: A Systematic Review and Network Meta-analysis" (PMEDICINE-D-20-06010R4) for consideration at PLOS Medicine.

I have discussed the paper with our academic editor and it was also seen again by one reviewer. I am pleased to tell you that, once the minor editorial and production issues are dealt with we expect to be able to accept the paper for publication in the journal.

[LINK]

We hope receive your revised manuscript within 1 week. Please email us (plosmedicine@plos.org) if you have any questions or concerns.

Please let me know if you have any questions, and we look forward to receiving the revised manuscript shortly.   

Sincerely,

Richard Turner, PhD

rturner@plos.org

Requests from Editors:

Our academic editor requested that you include an additional supplemental table for the results from the following statement: "When adjusted for proportion of study participants who had received ART, using meta-regression for network meta-analysis, the adjusted results for aggregated and individual regimens were very similar to the unadjusted results (supplemental table X)." (noting that PLOS journals do not allow "data not shown" statements).

Please check the risk difference quoted as "8.8" in the abstract, line 26, which may be quoted as "8.9" in table 5. 

At line 333, for example, please adapt p values to "p<0.001" unless there is a specific statistical justification for quoting smaller exact values. 

Please avoid "p<0.0000", e.g. at line 344.

At line 409, should that be "However, for TST/IGRA negative ..."?

At line 416, please make that "statistically significantly more effective ...".

In reference 6, should that be "N Engl J Med"?

Please revisit reference 64, where it appears that the author list needs trimming to 6 names, followed by "et al". Are you able to add full access details?

Comments from Reviewers:

*** Reviewer #2:

In response to the analysis requested by the academic editor I have no further suggestions to make. Since this is an essentially ecological analysis I am not too surprised that it did not reveal anything startling but it is still worth having if it forestalls negative comments.

Michael Dewey

***

[LINK]

---

## [Editor Report · Decision Letter 5]

18 Jul 2021

Dear Dr Menzies, 

On behalf of my colleagues and the Academic Editor, Dr Suthar, I am pleased to inform you that we have agreed to publish your manuscript "Tuberculosis Preventive Therapy for People Living with HIV: A Systematic Review and Network Meta-analysis" (PMEDICINE-D-20-06010R5) in PLOS Medicine.

PRESS

Sincerely, 

Richard Turner, PhD 

rturner@plos.org